# Mung bean seed classification based on multimodal features and Kepler-optimized stacking ensemble learning model

Shaozhong Song[1,2], Fengwei Leng[1], Ming Fang[1], Xiaofeng An[2]*, Yaxin Cai[3]

1 School of Artificial Intelligence, Changchun University of Science and Technology, Changchun, China,
2 School of Data Science and Artificial Intelligence, Jilin Engineering Normal University, Changchun,
China, 3 School of Computer Science and Technology, Changchun University of Science and Technology,
Changchun, China

* an.xf@jlenu.edu.cn

## Abstract

Accurate classification of mung bean seeds is essential for enhancing both their nutritional value and crop yields. However, current methods are limited, primarily due to the time-consuming and inaccurate classification process resulting from a lack of diverse dataset features. To overcome these challenges, this study develops a multimodal dataset that integrates Raman spectral features and image-based features through early fusion. Furthermore, the classification of mung bean seed varieties is achieved in a rapid, accurate, and non-destructive manner by optimizing a stacking ensemble learning model using the Kepler Optimization Algorithm (KOA). The multimodal dataset comprises 59 features, selected using the Competitive Adaptive Reweighted Sampling (CARS) method. Specifically, 44 key features are extracted from 700 Raman spectral data points, while 15 key features are derived from 43 image numerical features. The study also used the Kepler Optimization Algorithm to optimize the parameters of various machine learning models, including Decision Tree (DT), Support Vector Machine (SVM), k-Nearest Neighbor (KNN), Backpropagation Neural Network (BPNN), Random Forest (RF), and Gradient Boosting Decision Tree (GBDT). By constructing a stacking ensemble learning model, the research effectively leverages the strengths of multiple classifiers, thereby enhancing the overall classification performance. Experimental results demonstrate that the proposed method significantly improves mung bean seed classification accuracy, with the Kepler-optimized stacking ensemble model achieving an accuracy of 90.71%. This represents a 3.24% improvement over KOA-RF and a 1.59% improvement over KOA-GBDT. In comparison to baseline models, the proposed method proves to be more efficient. This study underscores the potential of combining multimodal features with a Kepler-optimized stacking ensemble learning model for mung bean seed classification. It highlights the role of advanced artificial intelligence techniques in

**Data availability statement:** The data used in this study are publicly available on the GitHub platform. The dataset can be accessed via the following link: https://github.com/781118954/multi-data. All relevant data are openly available to ensure the reproducibility of this study.

**Funding:** This study was supported by the Natural Science Foundation of Jilin Province in the form of a grant awarded to SS (No.20220101144JC) and the Natural Science Foundation of Jilin Province in the form of a salary for SS. The specific roles of this author are articulated in the 'author contributions' section.

**Competing interests:** The authors have declared that no competing interests exist.

agricultural production and provides valuable technical support for the precise classification of mung bean seeds.

---

## 1. Introduction

Mung bean has been cultivated as a globally significant crop for over 3,500 years [1], with cultivation in China dating back more than 2,000 years [2]. High-quality seeds are critical for sustaining both yield and product quality [3]. Seed yield, a key economic metric, is heavily influenced by cultivar selection [4], which further dictates the nutritional profile including oil, protein, and sugar content [5]. Accurate variety identification remains challenging due to the high degree of morphological similarity observed across different varieties.Traditional early classification of mung bean seeds relied on morphological traits, which were objective but inefficient [6]. Subsequent DNA fingerprinting methods enhanced efficiency but necessitated specialized laboratories and carried risks of seed damage [7]. Raman spectroscopy addresses the limitations of traditional methods by leveraging its rapid, non-destructive analytical capabilities [8]. This technique operates on the principle of inelastic light scattering and has been widely applied to plant variety identification and nutrient analysis [9,10], including assessing maize kernel viability [11], analyzing quinoa seed nutrients [12], and quantifying maize oil content [13]. While effective for evaluating inherent qualities, Raman spectroscopy exhibits limited utility in analyzing morphological characteristics. Computer vision employs image processing and analysis techniques to deliver precise and efficient solutions for plant phenotyping [14,15]. This technology predominantly relies on distinguishing features such as color, texture, and shape [16]. Notably, it has demonstrated successful applications in crop-weed identification [17], wheat variety identification [18], and maize seed identification [19]. However, its effectiveness hinges critically on the visual integrity of the target samples. Multimodal datasets, which integrate spectral and image features, have attracted considerable attention in classification and identification tasks. These datasets enable a more thorough and precise extraction of target object features.They have proven highly effective in applications such as rice and wheat seed variety classification [20], hyperspectral image classification [21], and maize seed variety classification [22].

Machine learning models have become key tools for classification tasks due to their powerful feature learning capabilities [23]. Compared with traditional methods such as decision trees (DT) [24], support vector machines (SVM) [25], and the k-nearest neighbors (KNN) [26], ensemble learning models integrate predictions from multiple base models to more effectively reduce bias and variance [27], thereby significantly improving prediction accuracy [28]. This approach has been successfully applied across multiple domains. It achieved 95.79% accuracy in crop leaf disease identification using an ensemble of VGG11, ResNet18, and MobileNet models [29]. In peanut yield prediction, random forest (RF) and gradient boosting decision tree (GBDT) models attained $R^2$ values of 0.93 and 0.88, respectively [30]. Additionally, an ensemble of enhanced AlexNet, GoogLeNet, ResNet50, and MobileNetV3

models demonstrated a 99.69% accuracy on a rice dataset [31]. Machine learning model performance is highly dependent on hyperparameter configurations, necessitating systematic hyperparameter tuning to improve predictive accuracy. Researchers often utilize meta-heuristic optimization algorithms to efficiently search for optimal parameter settings through iterative exploration of the solution space [32]. These algorithms prioritize heuristic evaluation strategies over reliance on explicit mathematical representations of objective functions or constraint conditions [33]. For instance, particle swarm optimization has been applied to enhance backpropagation neural networks and random forest models [34], while modified cuckoo search algorithms have been employed to optimize extreme learning machines [35]. Genetic algorithms and genetic dominance approaches have also been used to refine support vector machines [33].

The high similarity in morphological traits, genetic makeup, and growth stages across various mung bean seed varieties poses significant challenges for accurate and efficient identification. To address the knowledge gap in mung bean seed variety classification, this study employs a dual-strategy approach: constructing a multimodal dataset to characterize mung seed features and developing an advanced stacking ensemble learning model to enhance classification accuracy. The primary contributions of this paper are outlined below:

(1). This study addresses the limitations of explainable analysis in the field of mung bean seed classification.The Raman spectral features and image numerical features are utilized for the inherent qualities and phenotypic analysis of mung bean seeds, respectively.

(2). This study addresses the limitations of existing mung bean seed datasets in thoroughly and precisely extracting features.The Raman spectroscopy features and numerical image features are extracted separately, then a multimodal dataset is constructed using a front-end fusion approach.

(3). This study addresses the limitations of existing model parameter optimization algorithms. The key parameters of the base learner in the stacking ensemble learning model are optimized utilizing the Kepler optimization algorithm (KOA).

(4). This study addresses the limitations in the ensemble stacking model construction process of existing methods. Machine learning models are optimized to function as base learners, while the meta-learner model is selected through a combined optimization strategy.

## 2. Materials and methods

As illustrated in Fig 1, the classification process of mung bean seeds is organized into six distinct stages: 1. Screening the original mung bean seed samples; 2. Acquiring the Raman spectral features and image information features of the seeds, followed by appropriate preprocessing; 3. Extracting features from both the Raman spectra and image data; 4. Conducting early fusion of the extracted salient features to construct a multimodal dataset.; 5. Inputting the multimodal dataset into established stacking ensemble learning model; and 6. Outputting the classification results and conducting a data analysis.

### 2.1. Mung bean seed dataset

The mung bean seed samples used in this experiment were provided by the Research Institute of the Jilin Academy of Agricultural Sciences. These seeds were meticulously selected and are widely distributed in the market. As shown in Fig 2, seven varieties of mung bean seeds were chosen for data collection: Jilv No. 7 (A), Jilv No. 11 (B), Jilv No. 13 (C), Jilv No. 9 (D), Baolv No. 2013 (E), Jilv No. 10 (F), and Jilv No. 5 (G). For each variety, 100 seeds were randomly selected, resulting in a total of 700 data samples.

In the acquisition of Raman spectral features for mung bean seeds, this study employed a Hooke D100 near-infrared confocal Raman spectroscopy system, which was equipped with a 785 nm wavelength laser operating at 50 mW power. To capture subtle spectral variations, a precise sampling method was utilized. Raman intensities were recorded at $2cm^{-1}$

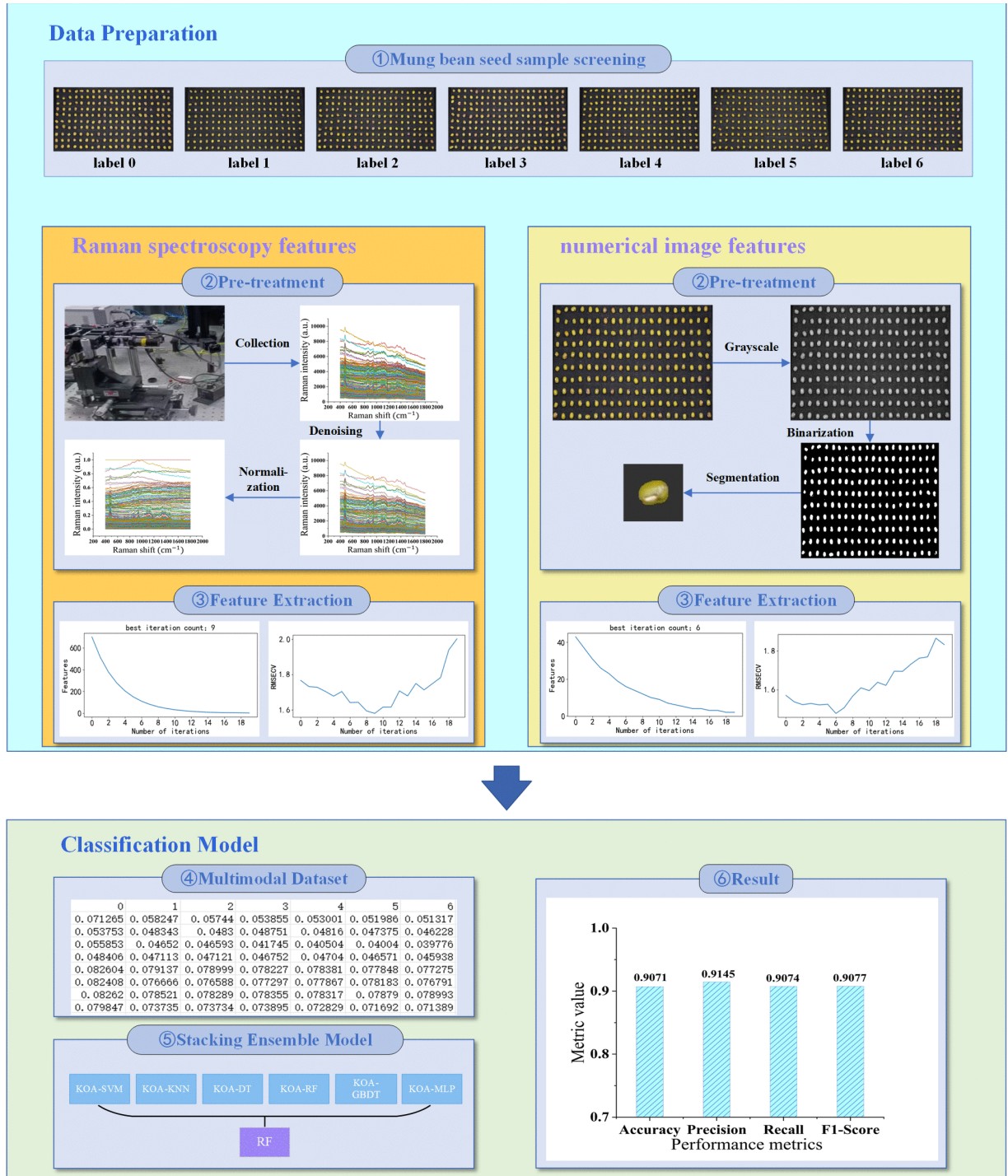

**Fig 1. The flowchart for mung bean seed classification based on multimodal features and Kepler-optimized stacking ensemble learning model.**

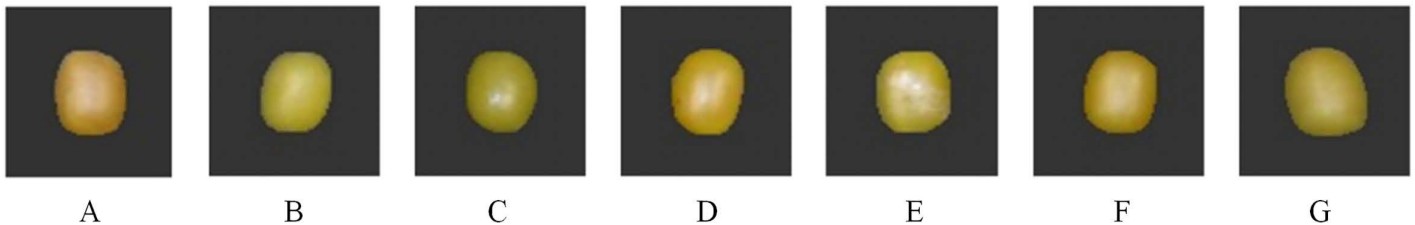

**Fig 2. Selected seven varieties of mung bean seeds.**

intervals across a Raman shift range from 400cm$^{-1}$ to 1800cm$^{-1}$, yielding a total of 700 distinct Raman spectral features. The Raman spectral features collection procedure is described in S1 Appendix.

In this study, mung bean seed images were captured using a Canon EOS 1500D camera, which has an effective resolution of 24.1 megapixels. To extract the image numerical features, the captured images were first converted to grayscale and then processed into binary images to facilitate effective separation of the seeds from the background. Using the binary image as a basis, the original images were accurately segmented, allowing for the extraction of individual mung bean seed images. From these segmented images, a total of 43 features were derived, including 18 geometric features, 7 texture features, and 18 color features. The image numerical features collection procedure is described in S2 Appendix.

## 2.2. Computational environment

The hardware specifications employed in this experiment are as follows: the processor is an Intel Core i5-12400F with a clock speed of 2.50 GHz, the system is equipped with 16 GB of RAM, and the graphics card utilized is an NVIDIA RTX 4060 Ti.

The software consisted of the Windows 11 operating system, Python 3.11.5, and the following primary libraries: scipy 1.11.1, scikit-learn 1.3.0, and mlxtend 0.23.1. Links to the packages and functions used in this study are provided at: https://github.com/781118954/multi-data.

## 2.3. Data preprocessing

Preprocessing operations were performed on the acquired raw Raman spectral features and image numerical features, respectively.

The raw Raman spectra features were first smoothed using a Savitzky-Golay (SG) filter, implemented via the savgol_filter function from the scipy.signal module, to reduce noise for subsequent analysis [36]. This function's performance depends on two key parameters: the window size, governing the degree of smoothing, and the polynomial order, influencing its ability to preserve fine spectral details. Given the intricate nature of the Raman spectra features, a window size of 15 and a polynomial order of 3 were chosen to balance effective noise reduction with retention of peak characteristics. Following this, max-min normalization was applied to standardize the spectral features onto a uniform scale, enabling meaningful comparative analysis. For the image numerical features, the max-min normalization method was applied to process the data. This step was necessary because the collected image numerical features encompass geometric characteristics, textural properties, and color attributes, which typically exhibit significantly different value ranges. Normalization ensures comparability among these heterogeneous features.

Following the aforementioned steps, all data were standardized and consistent, providing a reliable foundation for subsequent analysis.

## 2.4. CARS feature extraction

Competitive adaptive reweighted sampling (CARS) is grounded in Darwin's concept of natural selection, specifically the principle of "survival of the fittest," and enhances feature selection by mimicking the process of biological evolution [37]. The CARS was primarily implemented using the PLSRegression function from the sklearn.cross_decomposition library, with its core workflow illustrated in Fig 3.

CARS is distinguished by a two-phase feature extraction process, comprising an initial fast extraction stage followed by a more refined extraction stage. The objective of the fast extraction phase is to efficiently eliminate a substantial number of redundant and irrelevant features. During this phase, the exponential degradation function (EDF) is primarily employed to calculate the ratio of retained features and to discard those with minimal weights. For a dataset containing $m$ features with a sampling number $N$, the proportion of the number of features retained in the $i$th iteration is given in Eq. (1).

$$r_i = ae^{-ki} \tag{1}$$

To ensure that the first sampling includes all features and the $N$th sampling contains only two, the constants $a$ and $k$ are determined by Eq. (2):

$$a = \left(\frac{m}{2}\right)^{\frac{1}{N-1}} , \quad k = \frac{ln\left(\frac{m}{2}\right)}{N-1} \tag{2}$$

Following the initial screening of the EDF, a more refined selection of features is conducted using adaptive reweighted sampling (ARS). This technique involves weighted random sampling based on the current feature weights, where variables with higher weights are more likely to be retained, while those with lower weights are progressively discarded.

## 2.5. Stacking ensemble learning model

The stacking ensemble learning model enhances predictive performance by utilizing the predictions of multiple base learners as input features for the meta-learner. The principles of stacking ensemble learning model is shown in Fig 4. For a stacking ensemble learning model with $t$ base learners and one meta-learner, the input data is $x$, then the output $y$ can be expressed as Eq. (3):

$$y = f(h_1(x), h_2(x) \dots h_t(x)) \tag{3}$$

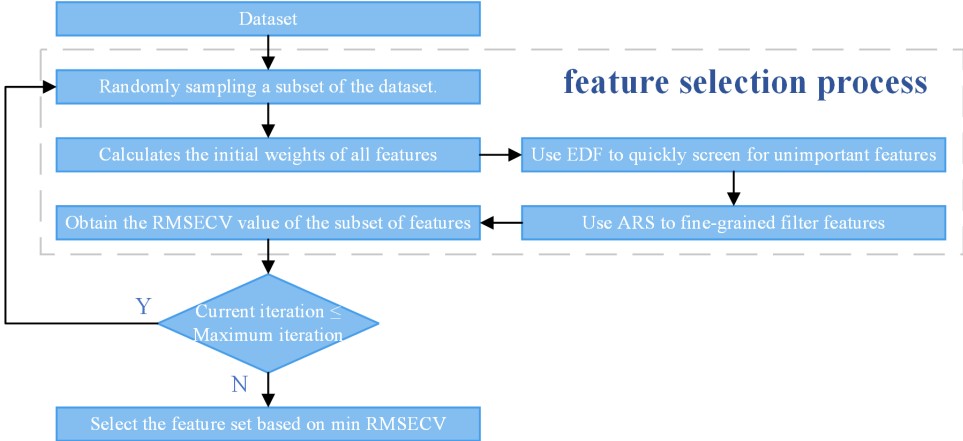

**Fig 3. Principles of CARS feature extraction.**

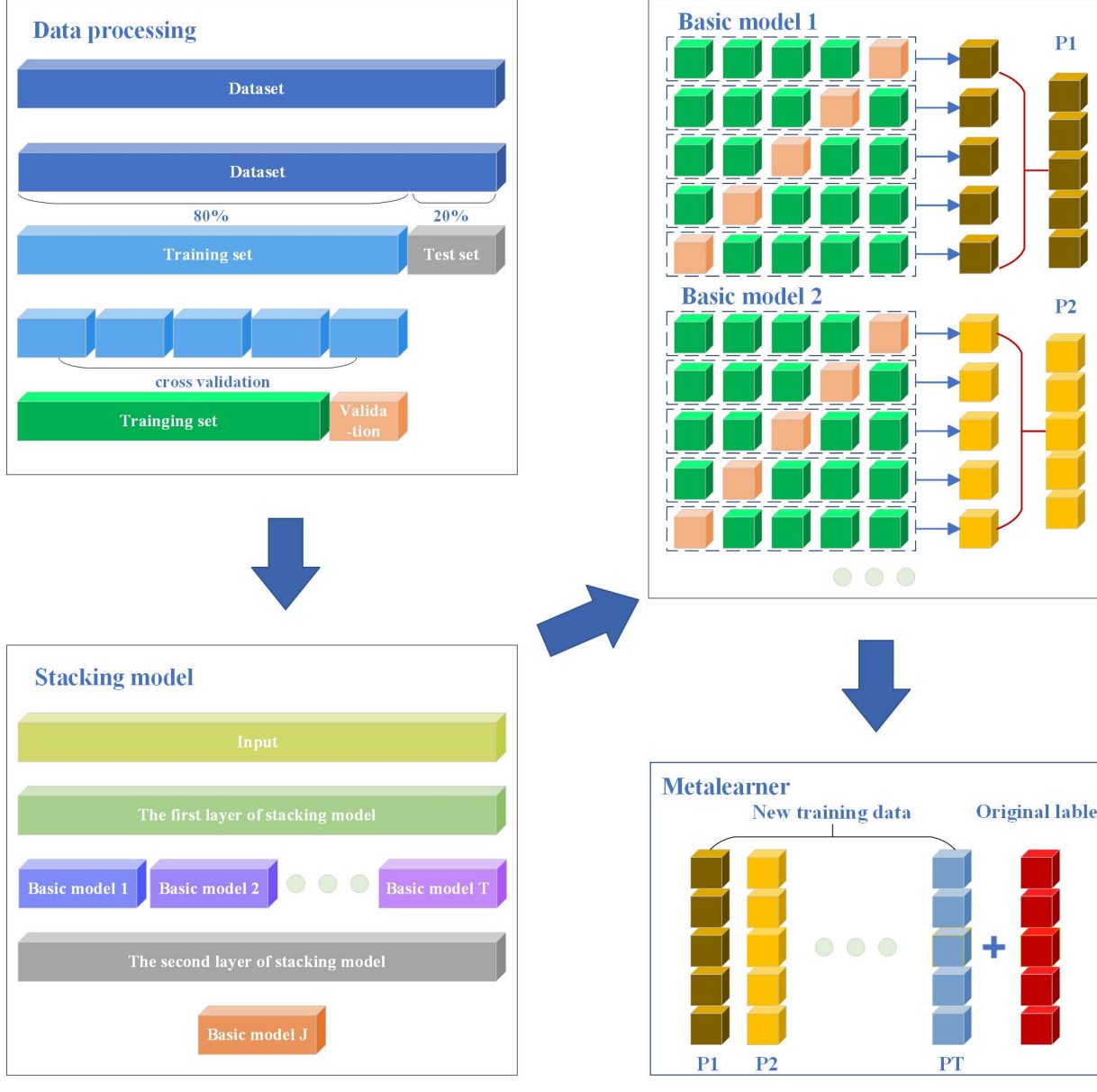

**Fig 4. Principles of stacking ensemble learning model.**

where $h_i(\cdot)$ represents the action of the base learner on the input and $f(\cdot)$ represents the action of the meta-learner on the input.

To enhance the generalization performance of the model and mitigate the risk of overfitting, stacking ensemble learning typically employs cross-validation to partition the dataset, which is subsequently used for model training. Assuming a training-to-test set ratio of 4:1, with k-fold cross-validation, the model's training process is outlined as follows:

(1). Data set division: The dataset is partitioned into a training set and a test set, with 80% of the data allocated to the training set and 20% to the test set. Subsequently, the training set is randomly divided into $k$ subsets of equal size. This step is primarily implemented using the train_test_split function from the sklearn.model_selection library.

(2). Training and prediction of base learner: Select one subset to serve as the validation set for the base learner, while the remaining $k$–1 subsets are used as the training set. For each base learner, train it using the designated training set. Subsequently, apply the trained base learner to the corresponding validation set to generate prediction results. This step is primarily implemented by initializing the classifiers parameter with base-learners within the StackingCVClassifier function from the mlxtend.classifier library.

(3). Cross-validation: This process is repeated for $k$ iterations, each time designating a different subset as the validation set. This method allows for the collection of prediction results, denoted as $P_1, P_2...P_k$, from all base learners across the $k$ iterations of cross-validation. The prediction result $P_i$ for an individual base learner is derived from the aggregation of predictions obtained during the $k$ cross-validation iterations.This step is primarily implemented by initializing the cv parameter within the StackingCVClassifier function from the mlxtend.classifier library.

(4). Constructing the secondary training set: The secondary training set is defined as the dataset formed by aggregating the base learners' predictions $P_1, P_2...P_k$ and their corresponding original labels, which is used for training the meta-learner.

(5). Training the meta-learner: The meta-learner is trained using a secondary training set. Its primary objective is to determine the most effective way to integrate the predictions from the first layer of the base learner, thereby enhancing the overall performance of the model.This step is primarily implemented by initializing the meta_classifier parameter with meta-learners within the StackingCVClassifier function from the mlxtend.classifier library.

## 2.6. Kepler optimization algorithm

The KOA is a novel meta-heuristic algorithm inspired by Kepler's three laws of planetary motion [38]. The principles of KOA is shown in Fig 5. The algorithm employs two strategies: randomly updating planetary positions and adjusting the distances of planets from the Sun. These strategies are designed to balance the algorithm's global and local search capabilities.

In the planetary position update strategy, the KOA emulates the motion of a planet as it gradually approaches the Sun in one segment of its elliptical orbit and recedes from the Sun in another segment. The position of the planet at iteration $t$+1 is denoted by $\vec{X}_i(t+1)$, which is mathematically represented by Eq. (4).

$$\vec{X}_i(t+1) = \vec{X}_i(t) + \mathsf{F} \times \vec{V}_i(t) + (F_{g_i}(t) + |r|) \times \vec{U} \times \left( \vec{X}_S(t) - \vec{X}_i(t) \right)$$

(4)

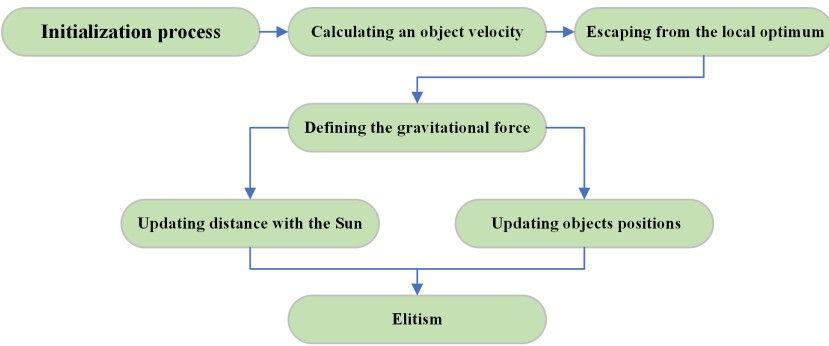

**Fig 5. Principles of Kepler optimization algorithm.**

$\overrightarrow{V}_i(t)$ is the velocity that moves the planet from position $\overrightarrow{X}_i(t)$ to $\overrightarrow{X}_i(t+1)$, and $F_{g_i}(t)$ is the gravitational force exerted by the Sun on the planet at position $\overrightarrow{X}_i(t)$.

In the strategy of updating the distance between the planet and the Sun, the KOA algorithm simulates the typical distance behavior that naturally changes over time between the Sun and the planet. At this point, the position $\overrightarrow{X}_i(t+1)$ can be obtained by Eq. (5).

$$\overrightarrow{X}_i(t+1) = \overrightarrow{X}_i(t) \times \overrightarrow{U}_1 + \left(1-\overrightarrow{U}_1\right) \times \left(\frac{\overrightarrow{X}_i(t) + \overrightarrow{X}_S + \overrightarrow{X}_a(t)}{3}\right) +$$

$$\left(1-\overrightarrow{U}_1\right) \times h \times \left(\frac{\overrightarrow{X}_i(t) + \overrightarrow{X}_S + \overrightarrow{X}_a(t)}{3} - \overrightarrow{X}_b(t)\right) \tag{5}$$

In this context, the regulating parameter $h$ plays a pivotal role. When its value is large, the exploration operation is employed to increase the distance between the planet's orbit and the Sun. Conversely, when the value is small, the Sun is positioned closer to the planets, and the exploitation operation is utilized to refine the region near the optimal solution.

## 2.7. Evaluation metrics

The performance for a target class A is evaluated based on four counts: true positives (TP) are samples from class A that are correctly identified; false positives (FP) are samples from other classes erroneously assigned to class A; true negatives (TN) are samples from other classes correctly identified as not belonging to class A; and false negatives (FN) are samples from class A that are incorrectly rejected.Based on the aforementioned fundamental statistics, the following core evaluation metrics can be defined:

(1). Accuracy measures the proportion of correctly classified instances relative to the entire dataset. Its formula is provided in Eq. (6)

$$accuracy = \frac{TP+TN}{TP+TN+FP+FN} \tag{6}$$

(2). Precision quantifies the proportion of true positives among all instances labeled as positive, reflecting the reliability of positive predictions. It is defined by Eq. (7).

$$precision = \frac{TP}{TP+FP} \tag{7}$$

(3). Recall, also known as sensitivity, represents the ratio of true positives to all actual positive instances. This metric evaluates the model's capacity to identify all relevant cases and is formulated in Eq. (8).

$$recall = \frac{TP}{TP+FN} \tag{8}$$

(4). F1-Score serves as the harmonic mean of precision and recall, offering a balanced evaluation metric particularly useful for datasets with class imbalance. Its calculation follows Eq. (9).

$$F1-Score = \frac{2 \times precision \times recall}{precision + recall} \tag{9}$$

# 3. Results and discussion

The stacking ensemble learning model adopted in this paper consists of two layers: base learners and a meta-learner. It fuses the prediction results of multiple base learners of different types and then conducts final integration through the meta-learner. By leveraging the complementarity among models, it captures the characteristics and patterns of data from multiple perspectives, effectively reducing errors and significantly improving the accuracy and generalization ability of prediction and classification. In view of this, selecting high – performing base learner models and meta-learner models is of great importance. In this paper, the KOA is first applied to optimize the parameters of six classification models, including SVM, DT, KNN, RF, MLP, and GBDT. The optimized models are then used as the base learners of the stacking ensemble learning model. Subsequently, the meta-learner model is selected through a combinatorial optimization approach, and finally, the stacking ensemble learning model used in this paper is constructed.

## 3.1. Feature extraction and result analysis

In this study, CARS was utilized to extract features from both the Raman spectral dataset and the image feature dataset, as depicted in Fig 6. In the context of CARS-based feature extraction, the features are conceptualized as "organisms," and the predictive performance of the model serves as an indicator of their "fitness." The model's prediction performance is treated as a measure of "adaptation." During each iteration, a subset of the extracted features is utilized in the Partial Least Squares Regression model, followed by cross-validation. The performance of each feature subset is evaluated by calculating its root mean square error of cross-validation (RMSECV), with the subset yielding the lowest RMSECV being selected as the optimal feature subset.

Fig 6 A illustrates the feature extraction process of the CARS method applied to the Raman spectral feature dataset. The process concludes after reaching the predefined maximum iteration count of 20. As depicted in Fig 6. A, the RMSECV reaches its minimum value of 1.58 at the 9th iteration. At this point, the number of selected features is 44, representing only 6.28% of the original feature set. The key features obtained during the extraction process of Raman spectral data are presented in Fig 7. A. Including 486cm$^{-1}$, 504cm$^{-1}$, 506cm$^{-1}$, 616cm$^{-1}$, 626cm$^{-1}$, 630cm$^{-1}$, 640cm$^{-1}$, 672cm$^{-1}$, 684cm$^{-1}$, 788cm$^{-1}$, 878cm$^{-1}$, 924cm$^{-1}$, 928cm$^{-1}$, 934cm$^{-1}$, 968cm$^{-1}$, 970cm$^{-1}$, 976cm$^{-1}$, 990cm$^{-1}$, 994cm$^{-1}$, 1012cm$^{-1}$, 1140cm$^{-1}$, 1150cm$^{-1}$, 1338cm$^{-1}$, 1342cm$^{-1}$, 1344cm$^{-1}$, 1352cm$^{-1}$, 1356cm$^{-1}$, 1444cm$^{-1}$, 1446cm$^{-1}$, 1484cm$^{-1}$, 1492cm$^{-1}$, 1500cm$^{-1}$, 1506cm$^{-1}$, 1518cm$^{-1}$, 1524cm$^{-1}$, 1590cm$^{-1}$, 1592cm$^{-1}$, 1714cm$^{-1}$, 1716cm$^{-1}$, 1722cm$^{-1}$, 1728cm$^{-1}$, 1732cm$^{-1}$, 1770cm$^{-1}$, 1772cm$^{-1}$.

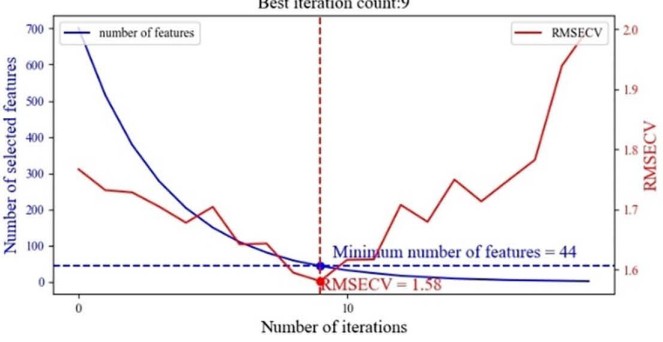
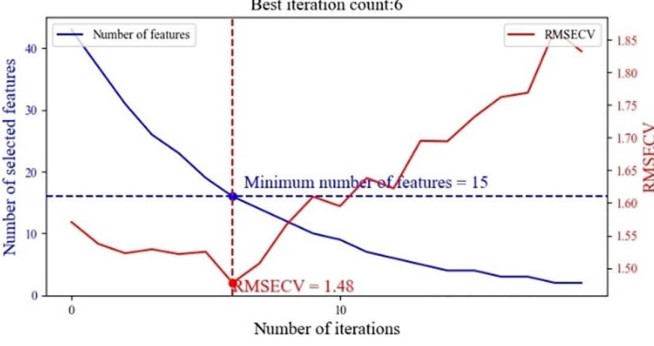

A CARS Raman spectroscopy process

B CARS Image Information Process

**Fig 6. CARS feature extraction iterative process.**

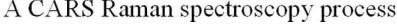

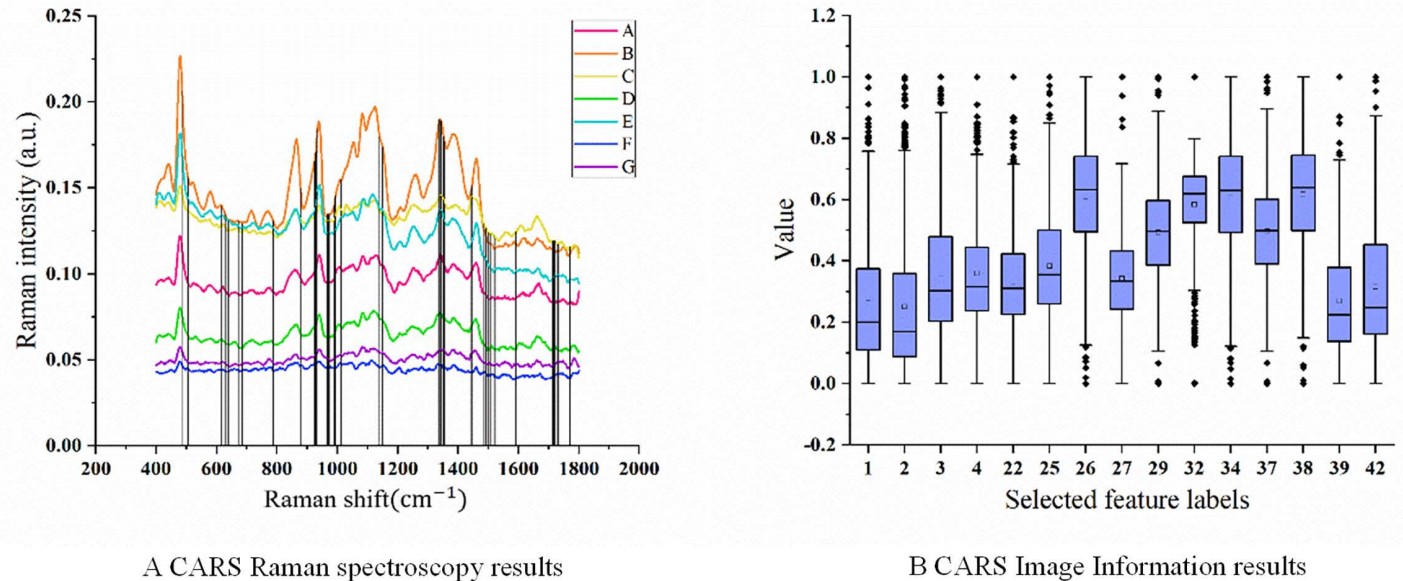

**Fig 7. Key features obtained during the CARS extraction process.**

Fig 6. B illustrates the feature extraction process of the CARS method applied to the image information feature dataset. Similarly, The RMSECV curve reaches its lowest point, with an RMSECV value of 1.48, at 6th iterations, where 15 features are selected. This represents 34.88% of the total number of initial features. The key features for image information extraction, as shown in Fig 7.B. Including 1. perimeter, 2. area, 3. long axis length, 4. short axis length, 22. grey scale variance product function, 25. vollath function, 26. r-mean in RGB space, 27. G-mean in RGB space, 29. R-variance in RGB space, 32. H-mean in HSV space, 34. V-mean in HSV space, 37. V-variance in HSV space, 38. L-mean in LAB space, 39. A-mean in LAB space, 42. A-variance in LAB space.

The Raman spectral features selected by the CARS algorithm in this work provide a statistically robust and chemically interpretable basis for classifying mung bean seeds. Peaks at $1444cm^{-1}$ and $1446cm^{-1}$ are unequivocally assigned to C-H bending vibrations, serving as a fingerprint for protein structure [39]. This critical assignment directly connects spectral data to the seed's biochemical makeup—specifically, protein content. Therefore, the high classification accuracy achieved in this study stems not from mere data fitting, but from the successful translation of molecular vibrational information into a discriminative basis for the machine learning model, thereby grounding the classification results in sound physicochemical principles.

### 3.2. Performance evaluation of multimodal datasets

Through early fusion, this study developed a multimodal feature dataset comprising a total of 59 features by combining 44 Raman spectral features with 15 features dataset. This fusion process is designed to integrate information from diverse data sources to enhance model performance and improve prediction accuracy. To investigate the impact of the constructed multimodal feature dataset on classification tasks, this study employs the classical DT model in machine learning to assess the classification performance across five datasets: the Raman spectral feature dataset, the image information feature dataset, the Raman spectral subset selected using CARS, the image information subset selected using CARS, and the multimodal feature dataset.

In this study, the dataset was divided into training and test sets at a ratio of 4:1, and performance evaluation results were obtained for five distinct datasets in the task of mung bean seed classification, as illustrated in Fig 8. When

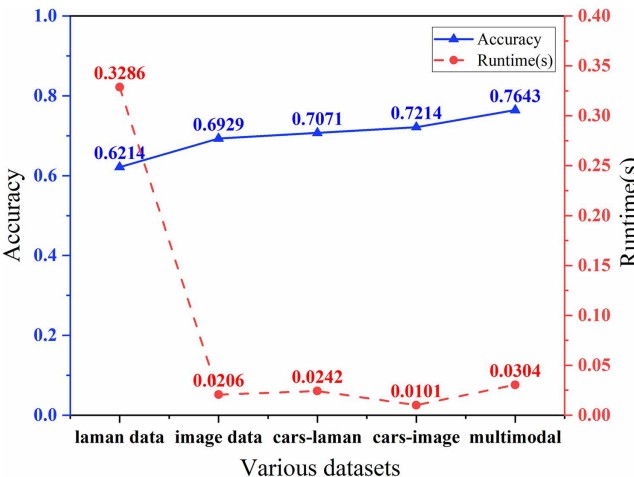

**Fig 8. Comparison of classification performance across various datasets.**

comparing the classification performance of the decision tree model applied to the Raman spectral feature dataset with that of the subset of Raman spectral features extracted using CARS, this study observed a 13.79% improvement in classification accuracy, while the runtime was substantially reduced to only 6.28% of the original. Similarly, after feature extraction on the image feature dataset, the classification accuracy increased by 4.11%, while the runtime decreased by 41.74%. Furthermore, when the decision tree model was applied to the multimodal dataset, the highest classification accuracy of 76.43% was achieved, with minimal increase in runtime.

Experimental results indicate that the multimodal dataset constructed in this study significantly outperforms single Raman spectral dataset in characterizing mung bean seeds [40]. This is attributed to the method's integration of Raman spectroscopy revealing internal chemical composition with image information reflecting external physical morphology, followed by effective redundancy removal through feature extraction, ultimately achieving a comprehensive, accurate, and concise characterization of seed features.

### 3.3. KOA optimizes the parameters of the base learner models

To ensure that the stacking ensemble learning model effectively captures the intrinsic features of the data from multiple perspectives, thereby enhancing predictive performance, this study carefully selected a set of base learners, drawing on the choices made by previous researchers [41,42]. The base learners chosen for the first layer of the ensemble model include: SVM, DT, KNN, RF, MLP and GBDT. To further improve the performance of these base learners, the study applied the KOA to optimize the model's parameters. After balancing search efficiency with accuracy, the population size was set to 10, and the maximum number of iterations was limited to 50. Additionally, the model's error rate was used as the fitness function. The parameter optimization ranges and the corresponding optimal values are presented in Table 1.

The KOA is capable of updating the best solution in each iteration, with the elite strategy ensuring that a superior candidate solution replaces the current best solution. This iterative process continues until the termination criteria, such as reaching the maximum number of iterations, are met. At this point, the algorithm concludes, and the current best solution is considered the optimal solution, marking the completion of the optimization process. The KOA demonstrates high efficiency in parameter optimization, as illustrated in Fig 9. In the parameter optimization task, the DT model attains a fitness function value of 0.2 in the first iteration. In contrast, the SVM model requires considerably more iterations to reach a fitness function value of 0.1214 in the 25th iteration. The KNN model achieves a fitness function value of 0.1571 by the

**Table 1. KOA optimize base learner parameters.**

| Basic model | Parameter Optimization Range | Optimal Parameters |
|---|---|---|
| DT | max_depth{1,100}, max_features{0.1,1} | max_depth = 13, max_features = 0.45 |
| SVM | C{1,1000}, gamma{0.1,1} | C = 828.92, gamma = 0.56 |
| KNN | n_neighbors{1,100}, p{1,5} | n_neighbors = 3, p = 2 |
| MLP | hidden_layer_sizes(k1,k2): k1{10,100}, k2{10,100} | hidden_layer_sizes(k1,k2): k1 = 96, k2 = 74 |
| RF | n_estimator{1,100}, max_depth{1,100}, max_features{0.1,1} | n_estimator = 92, max_depth = 78, max_features = 0.17 |
| GBDT | n_estimator{1,100}, max_depth{1,100}, max_features{0.1,1} | n_estimator = 84, max_depth = 20, max_features = 0.46 |

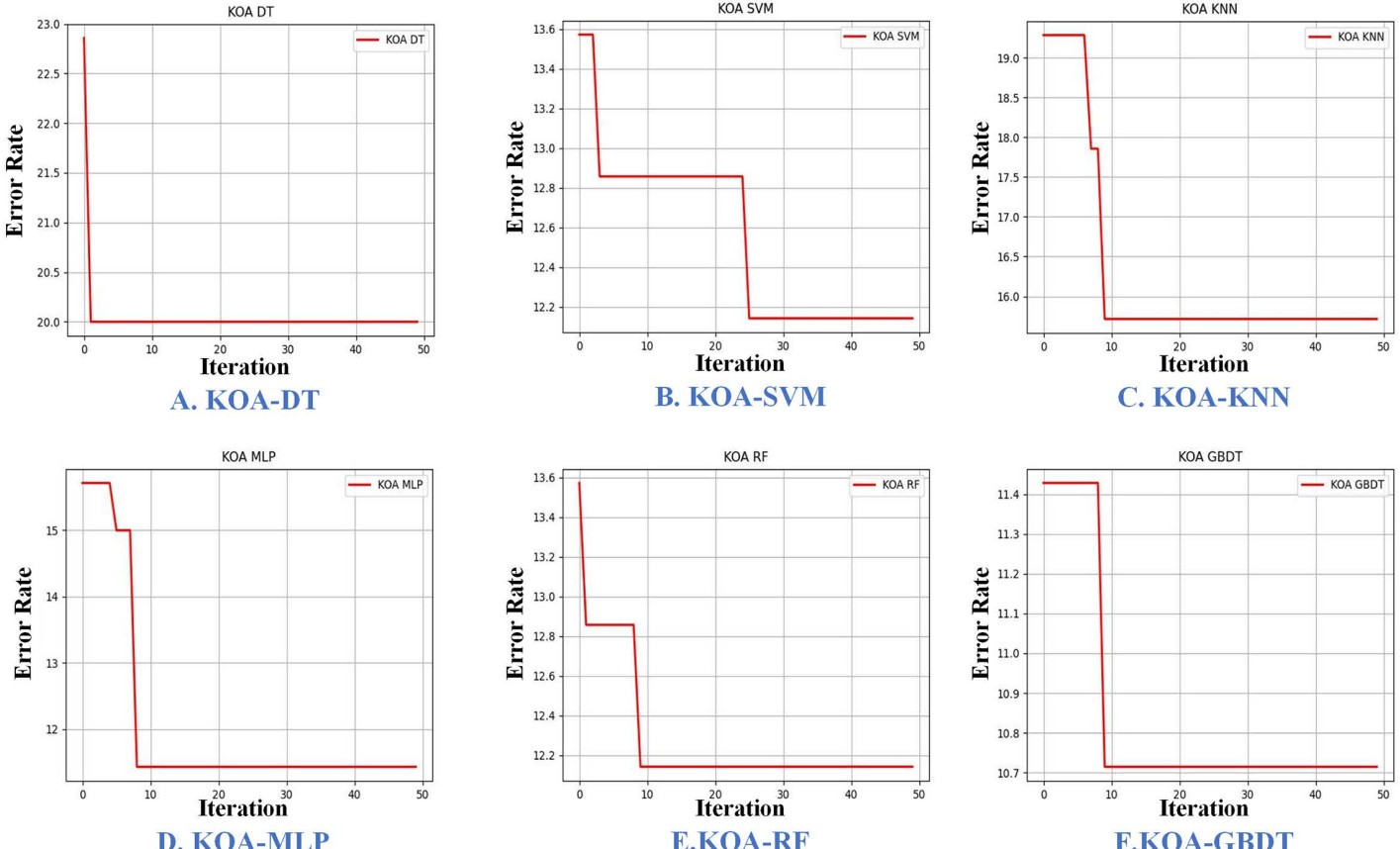

**Fig 9. The process of KOA optimizing base learning models.**

9th iteration, MLP model reaches a value of 0.1143 by the 8th iteration, RF model attains 0.1214 by the 9th iteration, and GBDT model reaches a value of 0.1071 by the 9th iteration. All of these models converge to the optimal parameter combinations relatively quickly, within 10 iterations.

The optimization of SVM parameters necessitates extensive iterations due to the strong coupling between regularization parameter C and gamma. These parameters collectively determine the algorithm's bias-variance trade-off: gamma

redefines the hypothesis space, which in turn shifts the optimal complexity level achievable by C [43]. This interdependence makes C's optimum contingent upon gamma's current value, transforming the optimization into a non-convex cooperative problem characterized by a response surface with abundant local optima and a globally optimal solution confined to a narrow gorge-like region, demanding highly synchronized parameter tuning.

The performance metrics of the six models—KOA-DT, KOA-SVM, KOA-KNN, KOA-MLP, KOA-RF, and KOA-GBDT—optimized using the KOA algorithm are presented in Table 2. Among these, the KOA-GBDT model demonstrates the best performance, achieving an accuracy of 0.8929, which represents a 2.63% improvement over the average accuracy of 0.8700 across all six models. Additionally, its precision is 0.9008, which is approximately 3.60% higher than the average precision of 0.8695; its recall is 0.8927, showing a 2.75% increase over the average recall of 0.8688; and its F1 score is 0.8923, which is 2.35% higher than the average F1 Score of 0.8718. These results underscore the effectiveness of the KOA algorithm in optimizing the parameters of machine learning models. By optimize these parameters, the model is able to more accurately capture data patterns during the learning process, thereby enhancing its predictive performance.

### 3.4. Stacking ensemble learning model construction and analysis

In constructing the stacking ensemble learning model, after optimizing the first layer of base learners, we employ combinatorial optimization to select the appropriate meta-learner. Experiments are conducted using DT, SVM, KNN, MLP, RF and GBDT as meta-learners. The resulting models are referred to as DT-stacking, SVM-stacking, KNN-stacking, MLP-stacking, RF-stacking, and GBDT-stacking, respectively. This approach aims to evaluate which meta-learner combination maximizes the classification performance of the model.

Table 3 demonstrates the classification performance of the six final stacking ensemble learning models constructed by this method. Among these models, the RF-stacking model has the best classification performance with 90.71% accuracy, 91.45% precision, 90.74% recall and 90.77% F1-Score. Compared to the KNN-stacking model, which has the second highest classification performance, RF-stacking is 3.24% higher in accuracy, 2.81% higher in precision, 2.60% higher in recall, and 3.02% higher in F1-Score.

**Table 2. Performance of models following KOA optimization.**

| Optimized model | Accuracy(%) | Precision(%) | Recall(%) | F1-Score(%) |
|---|---|---|---|---|
| KOA-DT | 0.8 | 0.7997 | 0.8176 | 0.8024 |
| KOA-SVM | 0.8786 | 0.8878 | 0.8818 | 0.8800 |
| KOA-KNN | 0.8429 | 0.8478 | 0.8510 | 0.8408 |
| KOA-MLP | 0.8857 | 0.8956 | 0.8894 | 0.8886 |
| KOA-RF | 0.8786 | 0.8860 | 0.8794 | 0.8766 |
| KOA-GBDT | 0.8929 | 0.9008 | 0.8927 | 0.8923 |

**Table 3. Performance of six stacking ensemble learning models.**

| Stacking model | Accuracy(%) | Precision(%) | Recall | F1-Score |
|---|---|---|---|---|
| DT-stacking | 0.8 | 0.7929 | 0.7921 | 0.7880 |
| SVM-stacking | 0.8714 | 0.8866 | 0.8772 | 0.8763 |
| KNN-stacking | 0.8786 | 0.8895 | 0.8844 | 0.8811 |
| MLP-stacking | 0.8714 | 0.8868 | 0.8732 | 0.8750 |
| **RF-stacking** | **0.9071** | **0.9145** | **0.9074** | **0.9077** |
| GBDT-stacking | 0.8714 | 0.8753 | 0.8707 | 0.8699 |

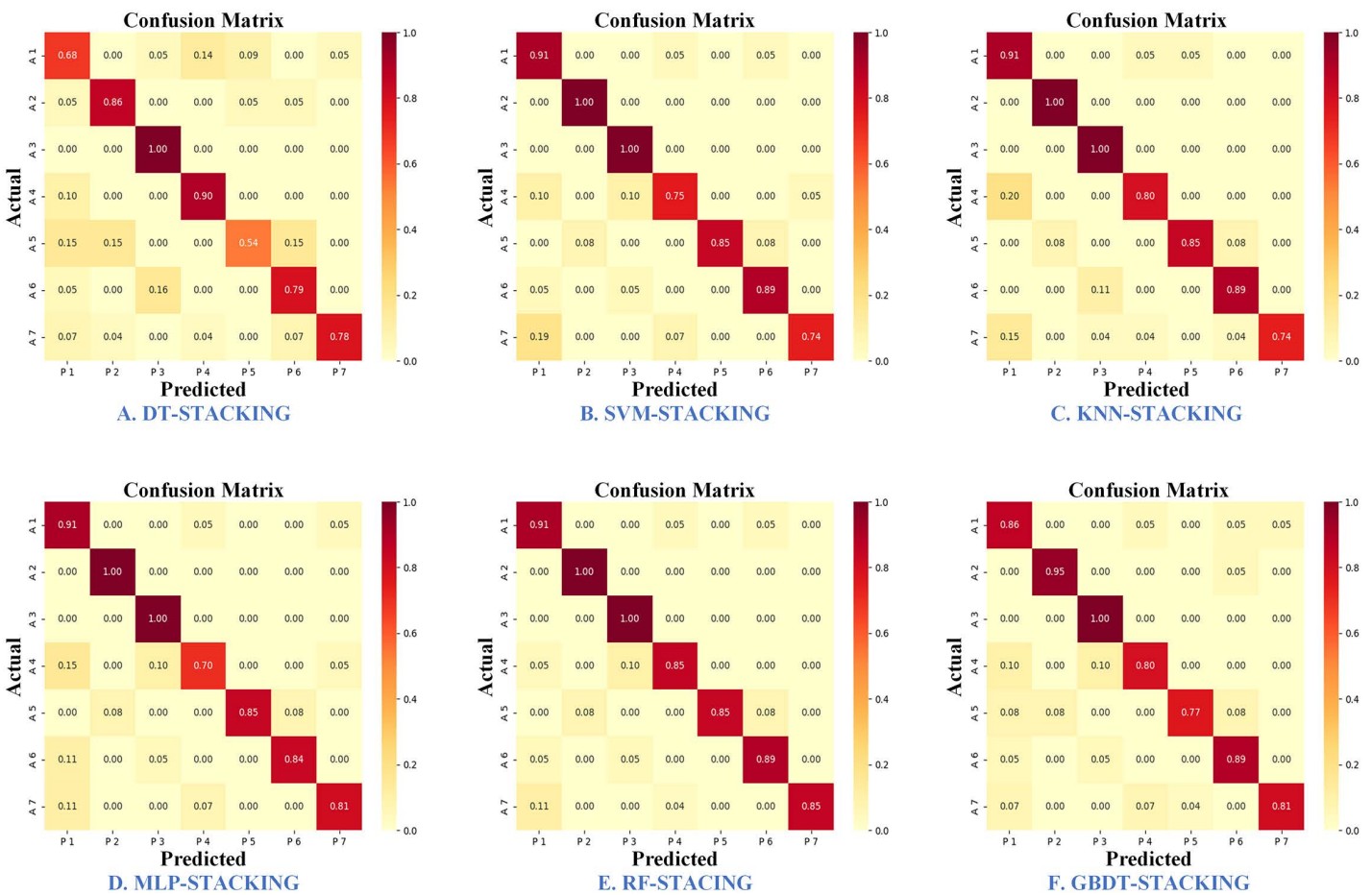

Fig 10 illustrates the confusion matrices of six stacking ensemble learning models, highlighting their performance in classifying mung bean seeds. Among these models, DT-stacking exhibits poor performance in classifying Baolv No. 2013 and Jilv No. 7. SVM-stacking struggles to differentiate between Jilv No. 9 and Jilv No. 5, while KNN-stacking shows weak identification accuracy for Jilv No. 5. MLP-stacking faces challenges in classifying Jilv No. 9, and GBDT-stacking also fails to effectively classify Baolv No. 2013. The classification of Jilv No. 9, Baolv No. 2013, and Jilv No. 5 proves particularly difficult across the models, with RF-stacking demonstrating the best overall classification performance.

Fig 11 illustrates the Receiver Operating Characteristic (ROC) curves for six stacking ensemble learning models, which are employed to evaluate the performance of the classifiers. The ROC curves are quantified by the Area Under the Curve (AUC), with higher AUC values indicating superior performance. In the context of multi-class classification, ROC curves are generated for each individual class to assess the model's ability to differentiate between them. The figure reveals that the DT-stacking model struggles to distinguish between Baolv No. 2013 and Jilv No. 7, while the Random Forest RF-stacking model demonstrates robust performance across all categories, achieving an average AUC value of 0.9886. Based on these results, we have selected RF model as the meta-learner in the stacking ensemble to optimize classification performance.

As shown in Fig 12, the RF-stacking model demonstrates a substantial performance advantage over the two classical integrated learning model, KOA-RF and KOA-GBDT, across all four key performance metrics: accuracy, precision,

**Fig 10. Confusion matrix for six stacking ensemble learning models.**

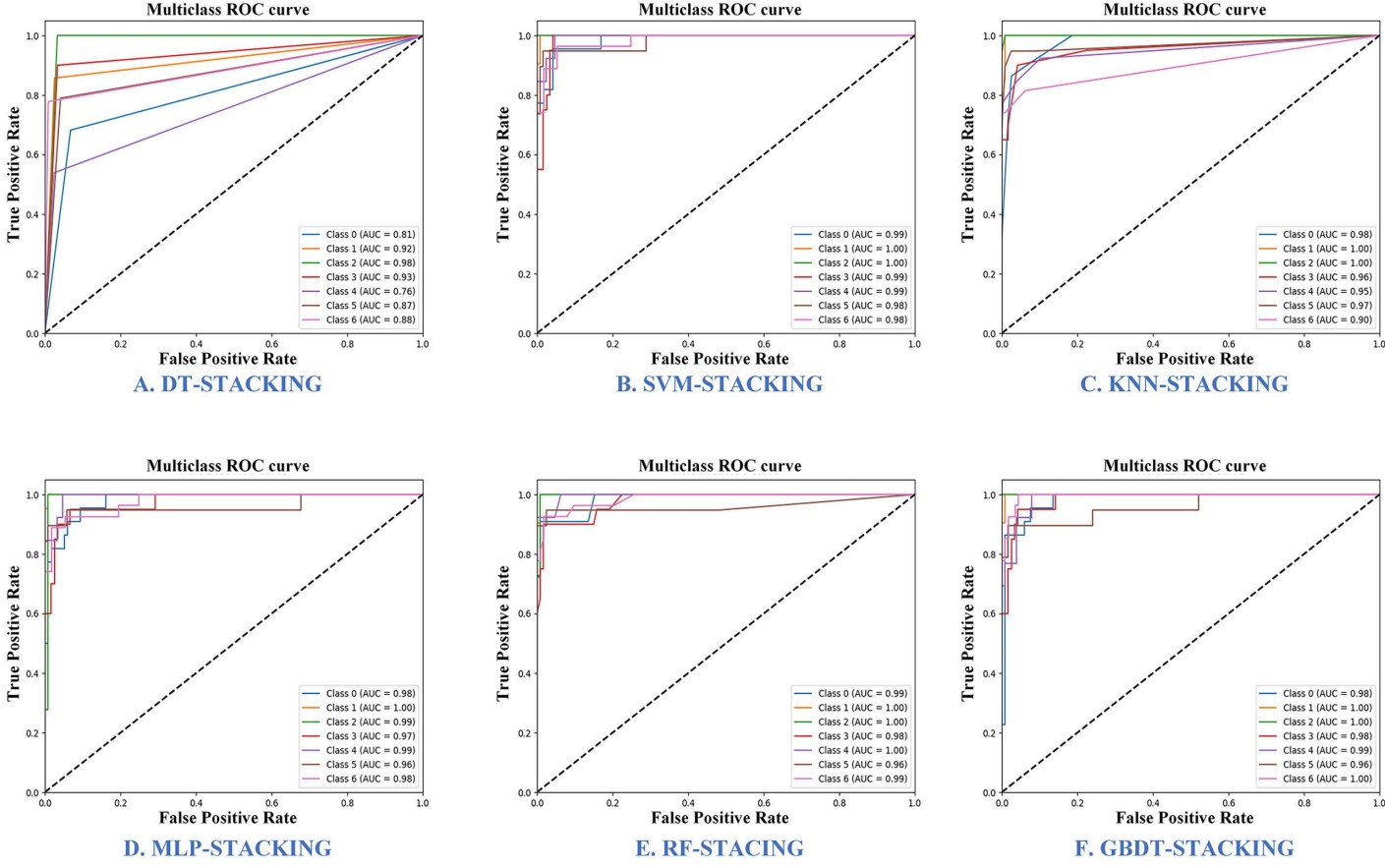

**Fig 11. ROC curves for six stacking ensemble learning models.**

recall, and F1-Score. Specifically, improvements of 3.24%, 3.22%, 3.18%, and 3.55% for accuracy, precision, recall, and F1-Score, respectively, are observed for the RF-stacking model compared to KOA-RF, and improvements of 1.59%, 1.52%, 1.65%, and 1.73% compared to KOA-GBDT.

Experimental results demonstrate that the proposed stacking ensemble learning model significantly outperforms individual RF and GBDT methods in mung bean seed classification tasks [44,45]. The performance enhancement is achieved through the integration of complementary characteristics from base learners. Specifically, RF exhibits powerful capability in capturing feature interactions, enabling comprehensive exploration of nonlinear relationships within the data, while GBDT demonstrates exceptional performance in modeling complex decision boundaries through efficient gradient optimization strategies. This synergistic interaction substantially enhances the model's generalization capability. Furthermore, at the architectural level, the stacking ensemble achieves superior bias-variance balance through its hierarchical framework, and employs meta-learners to intelligently integrate outputs from base models, thereby establishing a more equilibrium state between bias and variance.

## 4. Conclusion

This study introduces, for the first time, the stacking ensemble learning model based on multimodal features and the KOA, demonstrating strong performance in the classification of mung bean seeds. The experimental procedure and results are as follows: Initially, 44 key features are extracted from 700 Raman spectral features using the CARS feature extraction method,

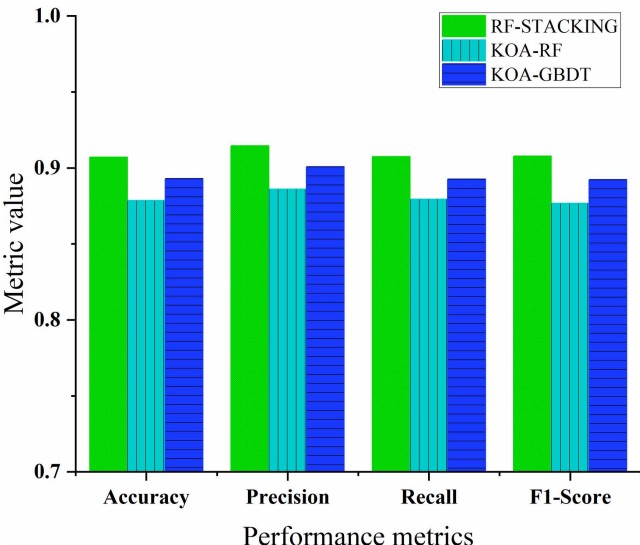

**Fig 12. Comparison of classification performance of different ensemble learning models.**

and 15 key features are extracted from 43 numerical image features. These 59 combined features form a multimodal dataset, which integrates information from multiple data sources while eliminating redundant features, thereby enhancing classification efficiency and accuracy. Subsequently, the KOA is employed to effectively search for optimal parameters for the base learner model, further improving classification performance. Finally, the optimized base learner model is used to construct a stacking ensemble learning model, and combinatorial optimization is applied to select the meta-learner model. This approach not only enhances accuracy but also mitigates the risk of overfitting and increases model flexibility.

The experimental results demonstrate that the RF-stacking ensemble learning model, constructed using KOA-DT, KOA-SVM, KOA-KNN, KOA-MLP, KOA-RF, and KOA-GBDT as base learners, with RF as the meta-learner, outperforms other baseline models in terms of accuracy, precision, recall, F1-score, and average AUC values, which are 0.9071, 0.9145, 0.9074, 0.9077, and 0.9886.It confirmed the effectiveness and practicality of the proposed method, offering a novel technical approach for the rapid, accurate, and non-destructive classification of mung bean seeds.

Future research will primarily focus on the following three directions. First, at the model architecture level, the introduction of lightweight deep learning models aims to overcome the limitations of traditional feature engineering and achieve efficient end-to-end feature learning. Second, at the data modality level, efforts will be dedicated to constructing a universal framework capable of flexibly integrating multi-source information, such as near-infrared spectroscopy and fluorescence spectroscopy, thereby transcending the current constraint of solely combining Raman spectral features and image numerical features. Finally, at the application scope level, the study plans to extend the model specifically designed for mung bean classification in this research into a universal intelligent detection solution applicable to various crops through transfer learning techniques.

## Supporting information

**S1 Appendix. Raman spectral data collection procedure.**
(JPG)

**S2 Appendix. Image features collection procedure.**
(JPG)

## Author contributions

**Data curation:** Fengwei Leng.

**Funding acquisition:** Shaozhong Song.

**Methodology:** Xiaofeng An, Shaozhong Song.

**Project administration:** Xiaofeng An, Shaozhong Song.

**Resources:** Shaozhong Song.

**Software:** Fengwei Leng.

**Supervision:** Shaozhong Song, Ming Fang.

**Validation:** Fengwei Leng, Ming Fang, Yaxin Cai.

**Visualization:** Fengwei Leng.

**Writing – original draft:** Fengwei Leng, Yaxin Cai.

**Writing – review & editing:** Xiaofeng An, Fengwei Leng, Ming Fang, Yaxin Cai.

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
