## [Editor Report · Decision Letter 0]

2 Apr 2025

Dear Dr. An,

Thank you for submitting your manuscript to PLOS ONE. After careful consideration, we feel that it has merit but does not fully meet PLOS ONE’s publication criteria as it currently stands. Therefore, we invite you to submit a revised version of the manuscript that addresses the points raised during the review process.

We look forward to receiving your revised manuscript.

Kind regards,

Dr Prakash Arumugam

Academic Editor

PLOS ONE

**Journal Requirements:**

1. When submitting your revision, we need you to address these additional requirements. Please ensure that your manuscript meets PLOS ONE's style requirements, including those for file naming. The PLOS ONE style templates can be found at https://journals.plos.org/plosone/s/file?id=wjVg/PLOSOne_formatting_sample_main_body.pdf and https://journals.plos.org/plosone/s/file?id=ba62/PLOSOne_formatting_sample_title_authors_affiliations.pdf 2. Please note that PLOS ONE has specific guidelines on code sharing for submissions in which author-generated code underpins the findings in the manuscript. In these cases, we expect all author-generated code to be made available without restrictions upon publication of the work. Please review our guidelines at https://journals.plos.org/plosone/s/materials-and-software-sharing#loc-sharing-code and ensure that your code is shared in a way that follows best practice and facilitates reproducibility and reuse. 3. Please include captions for your Supporting Information files at the end of your manuscript, and update any in-text citations to match accordingly. Please see our Supporting Information guidelines for more information: http://journals.plos.org/plosone/s/supporting-information.

**Additional Editor Comments:**

Dear author,

I recommend a Major Revision for your manuscript. Currently, the images are completely blurred, making it difficult to interpret the visual data accurately. Additionally, the modeling of the proposed method requires further clarification. It should be explained in a way that enhances understanding and benefits readers from various backgrounds. Please address these issues carefully to ensure that the manuscript meets the quality standards of PLOS ONE.

Thanks

Dr Prakash Arumugam

Academic Editor, PLOS ONE.

---

## [Author Response · Author response to Decision Letter 1]

14 Apr 2025

Point 1:When submitting your revision, we need you to address these additional requirements.Please ensure that your manuscript meets PLOS ONE's style requirements, including those for file naming.

Response 1:Thanks for your valuable comments.I formatted the article according to the journal's guidelines.

For example:Added line numbers to the title

Point 2:Please note that PLOS ONE has specific guidelines on code sharing for submissions in which author-generated code underpins the findings in the manuscript. In these cases, we expect all author-generated code to be made available without restrictions upon publication of the work. 

Response 2:Thank you for your valuable comments. In response, I have uploaded the complete code implementation with comprehensive annotations to GitHub. Both the source code and corresponding datasets are included in the same repository to ensure full reproducibility for the research community.

GitHub:https://github.com/781118954/multi-data

Point 3:Please include captions for your Supporting Information files at the end of your manuscript, and update any in-text citations to match accordingly.

Response 3:Thanks for your valuable comments.I have added the captions for the Supporting Information at the end of the manuscript and made the necessary in-text citations accordingly.

1.Captions for Supporting information

2.Corresponding in-text citations

Point 4(Additional Editor Comments):The images are completely blurred, making it difficult to interpret the visual data accurately. Additionally, the modeling of the proposed method requires further clarification. It should be explained in a way that enhances understanding and benefits readers from various backgrounds. Please address these issues carefully to ensure that the manuscript meets the quality standards of PLOS ONE.

Response 4:Thank you for your valuable feedback. Regarding the image resolution issue, I have updated the figure files using the Preflight Analysis and Conversion Engine (PACE) digital diagnostic tool provided by the journal, and they have been successfully re-uploaded. Additionally, the original composite image has been split into individual images and uploaded separately.

To address the need for further clarification of the modeling process, a detailed explanation has been incorporated between the theoretical framework and experimental sections to ensure comprehensibility for readers across diverse disciplinary backgrounds.

After verification

3. Results and discussion

The stacking ensemble learning model adopted in this paper consists of two layers: base learners and a meta-learner. It fuses the prediction results of multiple base learners of different types and then conducts final integration through the meta-learner. By leveraging the complementarity among models, it captures the characteristics and patterns of data from multiple perspectives, effectively reducing errors and significantly improving the accuracy and generalization ability of prediction and classification. In view of this, selecting high - performing base learner models and meta-learner models is of great importance. In this paper, the KOA is first applied to optimize the parameters of six classification models,including SVM, DT, KNN, RF, MLP, and AdaBoost. The optimized models are then used as the base learners of the stacking ensemble learning model. Subsequently, the meta-learner model is selected through a combinatorial optimization approach, and finally, the stacking ensemble learning model used in this paper is constructed.

The hardware specifications employed in this experiment are as follows: the processor is an Intel Core i5-12400F with a clock speed of 2.50 GHz, the system is equipped with 16 GB of RAM, and the graphics card utilized is an NVIDIA RTX 4060 Ti.

---

## [Decision Letter · Decision Letter 1]

14 Oct 2025

Dear Dr. An,

Thank you for submitting your manuscript to PLOS ONE. After careful consideration, we feel that it has merit but does not fully meet PLOS ONE’s publication criteria as it currently stands. Therefore, we invite you to submit a revised version of the manuscript that addresses the points raised during the review process.

We look forward to receiving your revised manuscript.

Kind regards,

Bappa Das

Academic Editor

PLOS ONE

Journal Requirements:

Additional Editor Comments (if provided):

The comments are provided within the attached PDF.

Reviewers' comments:

Reviewer's Responses to Questions

**Comments to the Author**

Reviewer #1: (No Response)

2. Is the manuscript technically sound, and do the data support the conclusions?

Reviewer #1: Yes

3. Has the statistical analysis been performed appropriately and rigorously?

Reviewer #1: Yes

4. Have the authors made all data underlying the findings in their manuscript fully available?

Reviewer #1: Yes

5. Is the manuscript presented in an intelligible fashion and written in standard English?

Reviewer #1: No

Reviewer #1: The introducton can be more chronological. The objectives needs to be written better. The execution of the algorithm and its environment is missing. The author needs to provide links to packages and functions used. The author can give link to the standard pdf used for the library used in the manuscript. The discussions and conclusions needs to be improved.

**Do you want your identity to be public for this peer review?** For information about this choice, including consent withdrawal, please see our Privacy Policy

Reviewer #1: **Yes: ** Ayushi Gupta

---

## [Author Response · Author response to Decision Letter 2]

9 Nov 2025

Response to Reviewers:

Thank you very much for taking the time to carefully review our manuscript and suggest valuable amendments. In response to the review opinions raised by reviewing experts, we first answered each question, and then revised the corresponding content in the paper according to the expert's comments (the revised part has been marked in red in the paper)

Point 1:Explain about Raman Spectroscopy and its paragraph from 78 to 88 line numbe should be here.

/The introducton can be more chronological.

Response 1:Thanks for your valuable comments.I have revised the sequence of the introduction section.:

“Raman spectroscopy addresses the limitations of traditional methods by leveraging its rapid, non-destructive analytical capabilities [8]. This technique operates on the principle of inelastic light scattering and has been widely applied to plant variety identification and nutrient analysis [9,10], including assessing maize kernel viability [11], analyzing quinoa seed nutrients [12], and quantifying maize oil content [13]. While effective for evaluating inherent qualities, Raman spectroscopy exhibits limited utility in analyzing morphological characteristics. Computer vision employs image processing and analysis techniques to deliver precise and efficient solutions for plant phenotyping [14,15]. This technology predominantly relies on distinguishing features such as color, texture, and shape [16]. Notably, it has demonstrated successful applications in crop-weed identification [17], wheat variety identification [18], and maize seed identification [19]. However, its effectiveness hinges critically on the visual integrity of the target samples.”

Point 2:write sentence structure like that " The study adresses the knowledge gap". The objective should not be mentioned as novel appproach or like that . /Write the objective in better way.

The objectives needs to be written better.

/Write this objectie in more simpler way

Response 2:Thank you for your valuable comments. In response, I have rewritten the research objectives using the writing structure you suggested.

“(1). This study addresses the limitations of explainable analysis in the field of mung bean seed classification.The Raman spectral features and image numerical features are utilized for the inherent qualities and phenotypic analysis of mung bean seeds, respectively.

(2). This study addresses the limitations of existing mung bean seed datasets in thoroughly and precisely extracting features.The Raman spectroscopy features and numerical image features are extracted separately, then a multimodal dataset is constructed using a front-end fusion approach.

(3). This study addresses the limitations of existing model parameter optimization algorithms. The key parameters of the base learner in the stacking ensemble learning model are optimized utilizing the Kepler optimization algorithm (KOA) .

(4). This study addresses the limitations in the ensemble stacking model construction process of existing methods. Machine learning models are optimized to function as base learners, while the meta-learner model is selected through a combined optimization strategy.”

Point 3:first write environment in which all these algorithm are processed with version of it used. Incluse the library used with its link to published pdf for help.

/The author needs to provide links to packages and functions used. The author can give link to the standard pdf used for the library used in the manuscript.

Response 3:Thanks for your valuable comments.I have provided the specifications of the hardware, software versions, and key libraries employed in this experiment, along with their relevant links.

“2.1Computational Environment

The hardware specifications employed in this experiment are as follows: the processor is an Intel Core i5-12400F with a clock speed of 2.50 GHz, the system is equipped with 16 GB of RAM, and the graphics card utilized is an NVIDIA RTX 4060 Ti.

The software consisted of the Windows 11 operating system, Python 3.11.5, and the following primary libraries: scipy 1.11.1, scikit-learn 1.3.0, and mlxtend 0.23.1. Links to the packages and functions used in this study are provided at: https://github.com/781118954/multi-data.”

Point 4:mention all the package and environmnet used for executing these package with the function to call for the algorithm

/write specific function with the packages used in each step of method

/The execution of the algorithm and its environment is missing.

/mention library with function used to execure for every major algorithm and accuracy analysis

Response 4: Thank you for your valuable feedback. I have specified the required libraries and functions for each individual algorithm.

“The raw Raman spectra features were first smoothed using a Savitzky-Golay (SG) filter, implemented via the savgol_filter function from the scipy.signal module, to reduce noise for subsequent analysis [37].”

“This step is primarily implemented using the train_test_split function from the sklearn.model_selection library.”

“This step is primarily implemented by initializing the classifiers parameter with base-learners within the StackingCVClassifier function from the mlxtend.classifier library.”

“This step is primarily implemented by initializing the cv parameter within the StackingCVClassifier function from the mlxtend.classifier library.”

“This step is primarily implemented by initializing the meta_classifier parameter with meta-learners within the StackingCVClassifier function from the mlxtend.classifier library.”

Point 5:no need to write "WE" ."The spectra was preprocessed"

Response 5: Thank you for your valuable feedback. I have revised the text using the sentence structures you suggested.

“Preprocessing operations were performed on the acquired raw Raman spectral features and image numerical features, respectively.”

Point 6:check all the equation,

Response 6:Thank you for your valuable feedback.I have double-checked all the formulas, ensuring they are properly formatted and use the Times New Roman font.

Point 7:There is only elaboration of results in whole section , the result significance with reference to previous research are discussed with the results.

/The discussions and conclusions needs to be improved

Response 7: Thank you for your valuable feedback. I have revised the text using the sentence structures you suggested.

(1).A qualitative analysis was performed on the extracted Raman spectral features.

(2) The multimodal dataset was compared with previous single Raman spectral datasets, and the reasons for the differences were analyzed.

(3) The phenomenon of high iteration counts during KOA-optimized SVM parameter search was analyzed.

(4) The performance of the stacked ensemble model constructed in this study was compared with other ensemble learning models, and the underlying reasons were discussed.

“The Raman spectral features selected by the CARS algorithm in this work provide a statistically robust and chemically interpretable basis for classifying mung bean seeds. Peaks at 1444cm−1 and 1446cm−1 are unequivocally assigned to C-H bending vibrations, serving as a fingerprint for protein structure [40]. This critical assignment directly connects spectral data to the seed's biochemical makeup—specifically, protein content. Therefore, the high classification accuracy achieved in this study stems not from mere data fitting, but from the successful translation of molecular vibrational information into a discriminative basis for the machine learning model, thereby grounding the classification results in sound physicochemical principles.”

“Experimental results indicate that the multimodal dataset constructed in this study significantly outperforms single Raman spectral dataset in characterizing mung bean seeds [41]. This is attributed to the method's integration of Raman spectroscopy revealing internal chemical composition with image information reflecting external physical morphology, followed by effective redundancy removal through feature extraction, ultimately achieving a comprehensive, accurate, and concise characterization of seed features.”

“The optimization of SVM parameters necessitates extensive iterations due to the strong coupling between regularization parameter C and gamma. These parameters collectively determine the algorithm's bias-variance trade-off: gamma redefines the hypothesis space, which in turn shifts the optimal complexity level achievable by C [44]. This interdependence makes C's optimum contingent upon gamma's current value, transforming the optimization into a non-convex cooperative problem characterized by a response surface with abundant local optima and a globally optimal solution confined to a narrow gorge-like region, demanding highly synchronized parameter tuning.”

“Experimental results demonstrate that the proposed stacking ensemble learning model significantly outperforms individual RF and GBDT methods in mung bean seed classification tasks [45,46]. The performance enhancement is achieved through the integration of complementary characteristics from base learners. Specifically, RF exhibits powerful capability in capturing feature interactions, enabling comprehensive exploration of nonlinear relationships within the data, while GBDT demonstrates exceptional performance in modeling complex decision boundaries through efficient gradient optimization strategies. This synergistic interaction substantially enhances the model's generalization capability. Furthermore, at the architectural level, the stacking ensemble achieves superior bias-variance balance through its hierarchical framework, and employs meta-learners to intelligently integrate outputs from base models, thereby establishing a more equilibrium state between bias and variance.”

Point 8:all the statistical and accuracy analysis statistical formula should be mentione in material section as a seprate heading

Response 8:Thank you for your valuable feedback.I have provided a detailed description of all statistical and accuracy analysis methods involved in this study.

“2.7Evaluation metrics

The performance for a target class A is evaluated based on four counts: true positives (TP) are samples from class A that are correctly identified; false positives (FP) are samples from other classes erroneously assigned to class A; true negatives (TN) are samples from other classes correctly identified as not belonging to class A; and false negatives (FN) are samples from class A that are incorrectly rejected.Based on the aforementioned fundamental statistics, the following core evaluation metrics can be defined:

(1). Accuracy measures the proportion of correctly classified instances relative to the entire dataset. Its formula is provided in Eq. (6)

(2). Precision quantifies the proportion of true positives among all instances labeled as positive, reflecting the reliability of positive predictions. It is defined by Eq. (7).

(3). Recall, also known as sensitivity, represents the ratio of true positives to all actual positive instances. This metric evaluates the model's capacity to identify all relevant cases and is formulated in Eq. (8).

(4). F1-Score (F1) serves as the harmonic mean of precision and recall, offering a balanced evaluation metric particularly useful for datasets with class imbalance. Its calculation follows Eq. (9). ”

Point 9:write short comings and challeneges faced which can be improved in fututre at the end of conclusion in last paragraph

Response 9:Thank you for your valuable feedback.I have outlined the limitations of this study and prospects for future research.

“Future research will primarily focus on the following three directions. First, at the model architecture level, the introduction of lightweight deep learning models aims to overcome the limitations of traditional feature engineering and achieve efficient end-to-end feature learning. Second, at the data modality level, efforts will be dedicated to constructing a universal framework capable of flexibly integrating multi-source information, such as near-infrared spectroscopy and fluorescence spectroscopy, thereby transcending the current constraint of solely combining Raman spectral features and image numerical features. Finally, at the application scope level, the study plans to extend the model specifically designed for mung bean classification in this research into a universal intelligent detection solution applicable to various crops through transfer learning techniques.”

---

## [Decision Letter · Decision Letter 2]

30 Nov 2025

Mung bean seed classification based on multimodal features and Kepler-optimized stacking ensemble learning model

PONE-D-24-59080R2

Dear Dr. An,

We’re pleased to inform you that your manuscript has been judged scientifically suitable for publication and will be formally accepted for publication once it meets all outstanding technical requirements.

Kind regards,

Bappa Das

Academic Editor

PLOS ONE

Additional Editor Comments (optional):

Reviewers' comments:

Reviewer's Responses to Questions

**Comments to the Author**

Reviewer #1: (No Response)

2. Is the manuscript technically sound, and do the data support the conclusions?

Reviewer #1: Yes

3. Has the statistical analysis been performed appropriately and rigorously?

Reviewer #1: Yes

4. Have the authors made all data underlying the findings in their manuscript fully available?

Reviewer #1: Yes

5. Is the manuscript presented in an intelligible fashion and written in standard English?

Reviewer #1: Yes

Reviewer #1: Address all the comments in the attachment. Improve all figure quality. Need special attention to the discussion of results.

**Do you want your identity to be public for this peer review?** For information about this choice, including consent withdrawal, please see our Privacy Policy

Reviewer #1: **Yes: ** Ayushi Gupta

---

## [Editor Report · Acceptance letter]

PONE-D-24-59080R2

PLOS One

Dear Dr. An,

I'm pleased to inform you that your manuscript has been deemed suitable for publication in PLOS One. Congratulations! Your manuscript is now being handed over to our production team.

Kind regards,

on behalf of

Dr. Bappa Das

Academic Editor

PLOS One